# Engineering of a FGM Interlayer to Reduce the Thermal Stresses Inside the PFCs

**Giacomo Dose [1,*], Selanna Roccella [2] and Francesco Romanelli [1]**

[1] Department of Industrial Engineering, University of Rome "Tor Vergata", Via del Politecnico 1, 00133 Rome, Italy

[2] Department of Fusion and Nuclear Safety Technology, ENEA, Frascati, 00044 Rome, Italy

\* Correspondence: giacomo.dose@uniroma2.it

**Featured Application: Thermomechanical design of a plasma-facing component with an engineered interlayer made with a functionally graded material, able to reduce the thermal stresses due to the mismatch of the coefficient of thermal expansion between the armor and the heat sink material.**

**Abstract:** A substantial contribution of the stresses that arise inside the Plasma-Facing Components (PFCs) when a heat load is applied is caused by the mismatch of the Coefficient of Thermal Expansion (CTE) between the armor, usually made of tungsten (W), and the heat sink. A potential way to reduce such contribution to the secondary stresses is the use of an interlayer made with a Functionally Graded Material (FGM), to be interposed between the two sub-components. By tailoring the W concentration in the volume of the FGM, one can engineer the CTE in such a way that the thermal stresses are reduced inside the PFC. To minimize and, theoretically, reduce to zero the stresses due to the CTE mismatch, the FGM should ensure kinematic continuity between the armor and the heat sink, in a configuration where they deform into exactly the shape they would assume if they were detached from each other. We will show how this condition occurs when the mean thermal strain of each sub-component is the same. This work provides a methodology to determine the thickness and the spatial concentration function of the FGM able to ensure the necessary kinematic continuity between the two sub-components subjected to a generic temperature field monotonously varying in the thickness, while remaining stress-free itself. A method for the stratification of such ideal FGM is also presented. Additionally, it will be shown that the bending of the PFC, if allowed by the kinematic boundary conditions, does not permit, at least generally, the coupling of the expansion of the armor and of the heat sink. As an example of our methodology, a study case of the thermomechanical design of a steel-based PFC with an engineered W/steel FGM interlayer is presented. In such an exercise, we show that our procedure of engineering a FGM interlayer is able to reduce the linearized secondary stress of more than 24% in the most critical section of the heat sink, satisfying all the design criteria.

**Keywords:** Plasma-Facing Components; functionally graded material; thermomechanical design

## 1. Introduction

One of the main functions of the Plasma-Facing Components (PFCs) in a fusion reactor is to exhaust the plasma heating power, which is the thermal power continuously fed to the fuel to sustain the fusion reactions. Due to the small heat exchange area, the deposited power density in steady state can reach tens of $MWm^{-2}$ in the regions where the plasma touches the wall (e.g., divertor and limiters), or be of the order of $1\ MWm^{-2}$ where the load is mainly due to radiation [1,2]. In their most general description, actively cooled PFCs are made of an armor and a heat sink. The first is the part of the component exposed directly to the plasma, made of a Plasma-Facing Material (PFM), having good compatibility with the plasma environment. Among the possible candidates, tungsten (W) is currently one of the

most promising choices [3]. The armor is bonded to the heat sink, which is the part of the PFCs where the coolant channels are located. For water-cooled PFCs, the heat sink material must be different from the PFM, since the commercially available W has a ductile-to-brittle transition temperature above 400 °C [4], which is incompatible with the coolant operative range. The fusion relevant heat sink materials are usually copper-based or iron-based [5]. When a thermal load is applied to a PFC, a temperature field builds up within the body. If such temperature field results in a congruent thermal strain field, meaning that it satisfies both the continuity equations in the medium and the external kinematic constraints at the boundary, then no stresses are generated. The simplest example is a uniform temperature field applied to an isostatically constrained body made of a homogeneous and isotropic material. If, conversely, the temperature leads to a non-congruent thermal strain field, then thermal stresses, also called secondary stresses, are generated. In other words, to prevent the solid domain from breaking apart, internal forces arise opposing the differential thermal expansion of the several parts of the body. A major contribution to the thermal stresses is due to the intrinsic bimetallic nature of the PFCs, which leads to a mismatch of the Coefficient of Thermal Expansion (CTE) between the PFM and the heat sink. Indeed, W has a CTE which is 3–4 times less than that of copper or iron [6]. Therefore, even if a homogeneous temperature field were applied to an isostatically constrained component, secondary stresses would arise since the heat sink would want to expand much more than the armor. In order to mitigate this contribution to the thermal stresses, a promising solution consists in adopting an interlayer made of a Functionally Graded Material (FGM) [7], to be interposed between the PFM and the heat sink. FGMs are composites which can vary in space their thermophysical properties, such as the CTE, by changing the local mixture ratio of its metallic components. The most relevant FGMs for fusion application are W/steel and W/copper [8,9]. By tailoring the W concentration in the volume of the FGM, one can engineer the thermal expansion of the component in such a way that the thermal stresses are reduced inside the PFC. In some works, different trends (linear, quadratic, exponential, etc.) of the spacial W content in the FGM interlayer, the so-called concentration function, were tested in a "trial-and-error" fashion in order to find the one minimizing the stresses [10]. Other studies [11–13], not specific to PFCs, optimize the FGM interlayer by minimizing numerically a cost function.

In this work, we instead propose a design methodology according to which the FGM concentration function is derived analytically from the governing equations of thermoelasticity. Specifically, the design of the interlayer is carried out by keeping constant the total strain that each sub-component (armor, FGM and heat sink) would experience if it were left free to expand, hypothetically detached from the rest. In other words, one engineers the interlayer, in its composition and thickness, in such a way that the armor would freely expand in the same fashion as the heat sink. A substantial difference of our approach from the mentioned optimization studies is that it allows to solve the thermoelastic problem for the separated armor and heat sink, which can be carried out analytically, reducing substantially the complexity from the formulation of a single body made of three different materials. After one computes separately the strain field experienced by the PFM and the heat sink, in the case that they both can deform freely due to the application of the temperature field, the FGM is then designed to make the two total strain fields match, while satisfying itself the same requirement. Such an optimization would allow to minimize, and ideally reduce to zero, the secondary stresses due to the mismatch of the CTE. In such a way, when the component is brought up to temperature the total deformation is equal to the one that its parts would experience if they were detached from one another. As explained in the following sections, this translates at least in a condition on the mean thermal strain of each sub-component. Another difference of our design approach from the optimization studies listed before, where the geometry of the problem is a given, is that only the thickness of the heat sink and of the armor are inputs. The first is derived such that the thermal stresses in the heat sink detached from the rest are below the allowable load. The latter is imposed by the plasma-wall interaction processes. The interlayer thickness is then found to be a key

engineering parameter for the reduction of the thermal stresses in the component, and will be determined such that no additional stresses will arise due to the CTE mismatch between the PFM and the heat sink.

The paper is organized in two main sections. In the first we derive the formulation for computing the total strain of freely supported bodies having a geometry relevant for the PFCs. Two cases will be described, both concerning solids having a thin thickness in the direction in which the thermal gradient is applied. The first consists in a hollow cylinder loaded homogeneously on the outer circumference, while the second is a plate loaded from one side. These examples with exact analytical solutions allow to understand the relevant engineering parameters. The substantial difference among the two instances is in the fact that the cylinder would not experience bending due to the axisymmetrical geometry and thermal boundary conditions, while the plate will have a curvature due to the monotonic nature of the temperature field. As we will see, if the bending occurs we cannot always find the matching of the "free" total strain of the PFM and of the heat sink, for all the possible choices of materials for the armor and the structural part of the PFC. In the second part of the paper, a design study is presented where the developed methodology is applied, specifically for the thermomechanical conceptual design of a steel-based PFC with an engineered W/steel FGM interlayer. Such component has a heat sink made of an AISI316L plate in which square channels for the water flow have been machined. In this design study, bending must be avoided for the above mentioned reason, through appropriate kinematic conditions at the boundary that an actual fixing system will have to best satisfy. In this section, we will also address the differences from the simple ideal case, such as the need of a FGM discretization and the inequality of the total strain components in the two main directions of the plate due to the presence of the coolant channels.

## 2. Analytical Methods for the FGM Engineering

In both cases that we will discuss, the design rationale is the same. First, the total strain experienced by the heat sink will be computed in the case where the body is left free to expand, ideally detached from the rest, due to the effect of the temperature field $T$. In both instances, $T = T(y)$ is a continuous monotonic function depending only on the thickness. We will assume that this scalar function is known a priori, in order to decouple the thermal from the thermomechanical problem. Actually, the temperature is not an arbitrary function, but instead is the solution of the Fourier equation, and it will depend on the boundary conditions, the FGM concentration and its thickness. Without loosing generality, we will assume that $T(y)$ is given. In the real case, one can easily come up with an iterative algorithm, converging on both the FGM thickness and concentration. However, this numerical effort is outside the scope of this work and instead we will focus on identifying the main parameters that drive the thermal stresses due to the CTE mismatch. In both cases that will be analyzed, one considers the Young's module $E$ and the Poisson's module $\nu$ of the materials to be constant with temperature. The approach used in this paper to derive the formulation of the total strain is analogous to the work found in [14,15] for one-dimensional structures. After the heat sink deformation is computed, it will then become the target value of the total strain. The armor is thus required to have the same strain field and the FGM thickness is determined to satisfy this very constraint. Finally, the ideal FGM composition is derived by imposing, everywhere in the interlayer, the thermal strain to be equal to the target strain. In such a way, it is directly derived from the constitutive relation ($\varepsilon = \varepsilon_T + \varepsilon_M$) that the FGM would be also stress-free since the mechanical strain $\varepsilon_M$ would be zero. The concentration function $C$ is correlated to the thermal strain $\varepsilon_T$ in the FGM, by adopting a linear mixture law for the secant CTE which is commonly used to model fusion-relevant FGMs [16,17]:

$$\alpha(T, C) = [\alpha_1(T) - \alpha_2(T)]C + \alpha_2(T) \tag{1}$$

where $\alpha$ is the CTE of the FGM, while $\alpha_1$ and $\alpha_2$ are the CTE of the two components of the FGM. The function $C$ is the volume concentration function of the material 1, in our case W.

By both using the definition of the thermal strain, $\varepsilon_T = \alpha(T, C)(T - T_0)$, in the constraints that we imposed in the proposed methodology and assuming a stress-free temperature $T_0$, the function $C$ is obtained. The modeling of the CTE using Equation (1) connects with already produced W/steel FGMs, showing good agreement with the measured CTE [18]. Furthermore, the CTE of fabricated W/copper FGMs [19], relevant for fusion applications, follow the linear mixture law, justifying the choice of such a model for the engineering of a FGM interlayer. Anyhow, from a mathematical point of view, our methodology is not limited to this specific model of the FGM CTE, but theoretically it could use any function $\alpha(T, C)$, as long as, at fixed $T$, the inverse function of $F_T(C) := \alpha(T, C)$ exists.

### 2.1. Hollow Cylinder with Thin Thickness and a Homogeneous Thermal Load

The constitutive equations in cylindrical coordinates $(r,t,z)$, for a 3D cylindrical hollow body having a small thickness $\delta$ and made of a homogeneous and isotropic material, are obtained by considering the radial normal stress to be negligible compared to the other components of the stress tensor ($\sigma_r \sim 0$):

$$\begin{cases} \varepsilon_t &= \frac{1}{E}(\sigma_t - \nu\sigma_z) + \alpha(T - T_0) \\ \varepsilon_z &= \frac{1}{E}(\sigma_z - \nu\sigma_t) + \alpha(T - T_0) \end{cases} \tag{2}$$

where $\varepsilon_i$ are the total strain components in the principal directions and $\sigma_i$ the principal stresses. The fact that $\delta$ is small compared to the other dimensions of the body, together with the lack of bending due to the axisymmetric geometry and temperature field, implies that the $\varepsilon_i$ can be considered constant over the thickness. Moreover, since the temperature field is a function of the cylinder thickness only, the principal strains do not vary with $t$ and $z$. Additionally, in absence of external forces, the integral over the thickness of all the principal stresses is zero. Therefore, by integrating Equation (2) over the thickness one finds:

$$\varepsilon_t = \varepsilon_z = \frac{1}{\delta} \int_0^{\delta} \alpha(T - T_0) dy \tag{3}$$

In this case, $y = 0$ at the inner radius and $y = \delta$ at the outer radius of the hollow body. If the tangential strain can be considered constant along the thickness, the congruence equation coming from the cylindrical geometry ensures that also the radial strain is constant and equal to $\varepsilon_r = \frac{d}{dr}(\varepsilon_t r) = \varepsilon_t$. As a result, the total strain components of the body are all equal to the mean thermal strain, $\varepsilon_{T,m} = \frac{1}{\delta} \int_0^{\delta} \alpha(T - T_0) dy$. This means that the cylinder, due to the thermal expansion generated by the temperature field $T$, deforms into a homothetically bigger cylinder having all its linear dimensions increased by a ratio $(1 + \varepsilon_{T,m})$. In the special case where $\alpha$ does not vary with the temperature, the cylinder deforms into the larger cylinder exactly the size it would have if heated uniformly to its average temperature. Therefore, the heat sink would experience such deformation if it were detached from the rest of the component. Using the same approach, one obtains that also the thin cylindrical armor would expand freely in this fashion, but with the substantial difference that, due to the dissimilar CTE and local temperature, the homotetic ratio will be different. This is in fact the reason why, once we bond the PFM to the heat sink, additional stresses are generated. However, if all the sub-components would expand in the same exact manner, this contribution to stresses would be eliminated. To achieve so, the mean thermal strain of the two bodies must be equal:

$$\varepsilon_{T,m1} = \varepsilon_{T,m2}$$
$$\frac{1}{\delta_1} \int_{\delta_2}^{\delta_1 + \delta_2} \alpha_1(T - T_0) dy = \frac{1}{\delta_2} \int_0^{\delta_2} \alpha_2(T - T_0) dy \tag{4}$$

where the subscript 1 refers to the PFM and 2 to the heat sink material. In the simplified case of linear thermoelasticity, for which $\alpha_i$ are constants, this equation translates in a condition on the mean temperatures:

$$T_{m1} = \frac{\alpha_2}{\alpha_1} T_{m2} + \left(1 - \frac{\alpha_2}{\alpha_1}\right) T_0 \tag{5}$$

In other words, if the W would be hot enough to match the heat sink's expansion, no stresses due to the CTE mismatch will be generated. A straightforward solution to satisfy such condition is to increase the armor thickness above the minimum requirement coming from the sputtering erosion. One must in this case solve Equation (4) for $\delta_1$. However, while reaching our objective, a thicker PFM would also increase the contribution to thermal stresses which are due to the temperature gradient. To clarify this aspect, we will consider for now a case where the CTE is temperature independent, $\alpha = \text{cost}$. We start by decomposing the monotone function $T(y)$ in thickness into a constant part $T_m = \frac{1}{\delta} \int_0^\delta T dy$, and a thickness-dependent term $T_1(y)$, whose integral over the thickness is zero. Therefore, we can write:

$$T(y') = T_m + T_1(y') \tag{6}$$

$y'$ is a new abscissa along the thickness where by definition $T(y' = 0) = T_m$. Let us now consider an elementary segment such as the one in Figure 1, and we consider it as if it were detached from the rest of the hollow cylinder. The thermal strain due to the temperature field $T(y')$ can also be decomposed into two terms: the first is a homothetic expansion $\varepsilon_{T,m} = \alpha(T_m - T_0)$ which causes an increase in the mean radius of the cylinder, plus a term depending on $T_1(y')$ that does not modify the volume of the cylinder but would distort the elementary element as shown in Figure 1. While the first contribution of the thermal deformation does not generate stresses as it automatically satisfies the continuity equations, the second tends to deform the segment in a non-congruent way and tangential stresses arise to "bring the fibers back" along the radii in such a way that the elementary segment satisfies the cylindrical symmetry. Such stresses are proportional to the thermal strain that they are opposing:

$$\sigma_t = -\frac{E\alpha}{1-\nu} T_1(y') \tag{7}$$

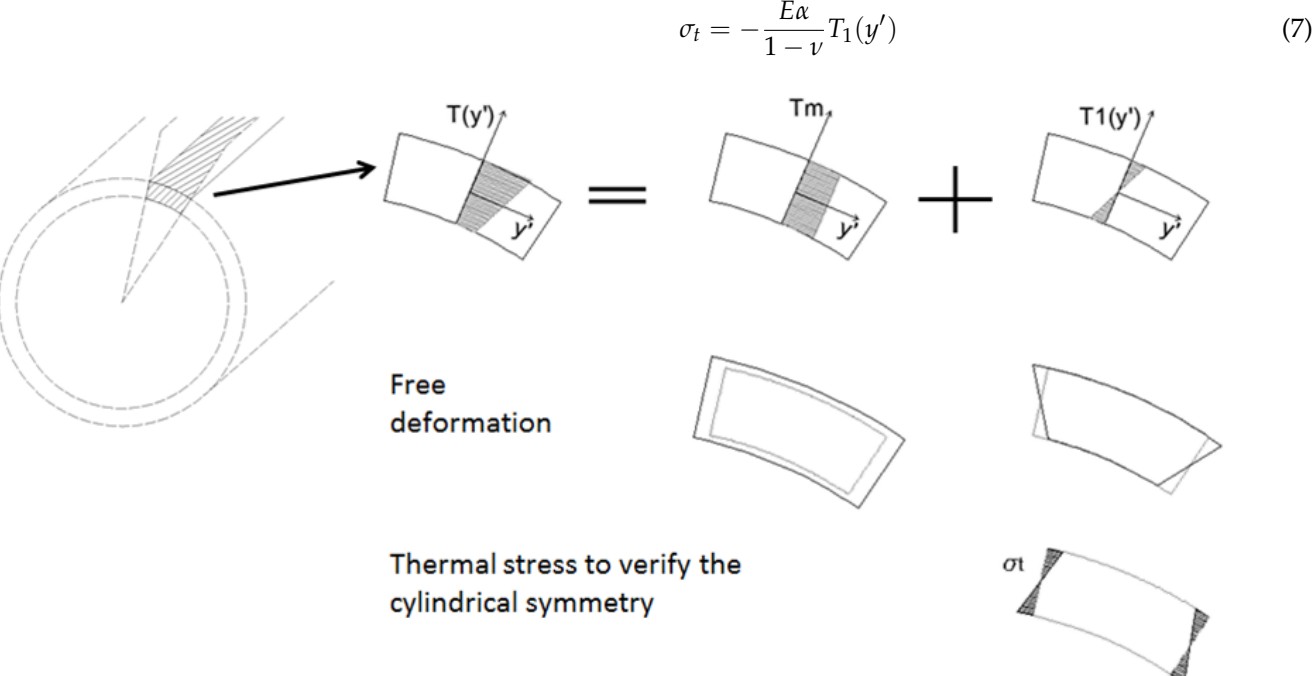

**Figure 1.** Deformation of an elementary segment of the hollow cylinder due to the temperature $T$. In the case of a temperature independent CTE, the thermal field can be decomposed in a constant term $T_m$ that generates a stress-free homotetic deformation, and a part $T_1(y')$ that generates the thermal stresses to satisfy the cylindrical symmetry.

If $T(y')$ were linear, $T_1(y')$ would be equal to $(T_{\max} - T_m)\frac{2y'}{\delta}$, where $T_{\max}$ is the maximum value of the temperature field. Therefore, the highest is the excursion between the maximum temperature value to the mean temperature, the more intense are the stresses. Such results can be generalized to the case with a temperature dependent CTE by decomposing the thermal strain field $\varepsilon_T(y)$ in two terms, with an analogous procedure as previously performed in the linear case to the field $T(y)$. The first contribution leads to a homotetic stress-free deformation, while the second is the one that generates the thermal stresses. Following the same approach, one obtains that in a general case the stresses depend on the quantity $(\varepsilon_{T,\max} - \varepsilon_{T,m})$, where $\varepsilon_{T,\max} = \alpha(T_{\max})(T_{\max} - T_0)$. Therefore, increasing $\delta_1$ would reduce the contribution generated by the CTE mismatch, but would increase as well the loads due to the temperature gradient by bringing $T_{\max}$ up. Moreover, it would be best to leave the armor thickness as a parameter determined only by the constraints due to the processes of plasma-wall interaction (PWI), such as the before mentioned erosion. To achieve a hot W layer without changing its size, one computes accordingly the thickness $\delta_3$ of an interlayer, put between the PFM and the heat sink. The FGM geometry must then satisfy:

$$\varepsilon_{T,m1} = \varepsilon_{T,m2}$$
$$\frac{1}{\delta_1} \int_{\delta_2+\delta_3}^{\delta_1+\delta_2+\delta_3} \alpha_1(T - T_0)dy = \frac{1}{\delta_2} \int_0^{\delta_2} \alpha_2(T - T_0)dy \tag{8}$$

In such a Equation, $\delta_1$ is imposed by the PWI processes, $\delta_2$ is an input pre-determined in such a way that the stresses due to the temperature gradient in the heat sink are below the maximum allowable value, and $\delta_3$ is the only unknown. The thickness of the interlayer is therefore a key engineering parameter to couple the total strain of the PFM to the one of the heat sink. The last step for the design of the interlayer is to determine the FGM concentration. The FGM must itself satisfy the condition on its mean thermal strain to be equal to the one of the other sub-components, otherwise the effort of eliminating the stress due to the CTE mismatch would be in vain:

$$\frac{1}{\delta_3} \int_{\delta_2}^{\delta_2+\delta_3} \alpha(T, C)(T - T_0)dy = \varepsilon_{T,m3} = \varepsilon_0 \tag{9}$$

where $\varepsilon_0$ is the target value of the total strain, defined by the heat sink expansion ($\varepsilon_0 = \varepsilon_{T,m1} = \varepsilon_{T,m2} = \varepsilon_{T,m3}$). From the infinite set of $\alpha$, solutions of Equation (9), one chooses the element for which the thermal strain of the FGM is equal to $\varepsilon_0$ everywhere in the interlayer thickness:

$$\alpha(T, C)(T - T_0) = \varepsilon_0 \tag{10}$$

In this way, one has tailored the thermal expansion in the FGM so that no thermal stresses arise in the material. In fact, as the thermal strain field is a constant, it automatically satisfies the congruence equations in the whole solid domain. Therefore, even the contribution due to the thermal gradient vanishes, resulting in a stress-free FGM. Combining Equations (1) and (10), the FGM concentration is derived as:

$$C = \frac{1}{\alpha_1(T) - \alpha_2(T)} \left[ \frac{\varepsilon_0}{T - T_0} - \alpha_2(T) \right] \tag{11}$$

Finally, one obtains the ideal FGM concentration, resulting in the minimization of the thermal stresses in the PFC. The armor and heat sink will be subjected only to the loads due to the thermal gradient, while the interlayer will be stress free. One notices, that $C$ is not defined for $T = T_0$. Therefore, one imposes that inside the FGM the temperature field must have values either strictly greater or strictly lower than the stress-free temperature. The results of a computation using the methodology we described are reported in Figure 2, that shows the trend over the cylinder thickness of the main parameters. The study has been carried out having a 1-mm-thick AISI316L cylindrical heat sink and 2 mm of W armor.

The FGM is developed as a mixture of these two materials. To carry out the computation, the temperature function $T(y)$ must be given. In our case we use:

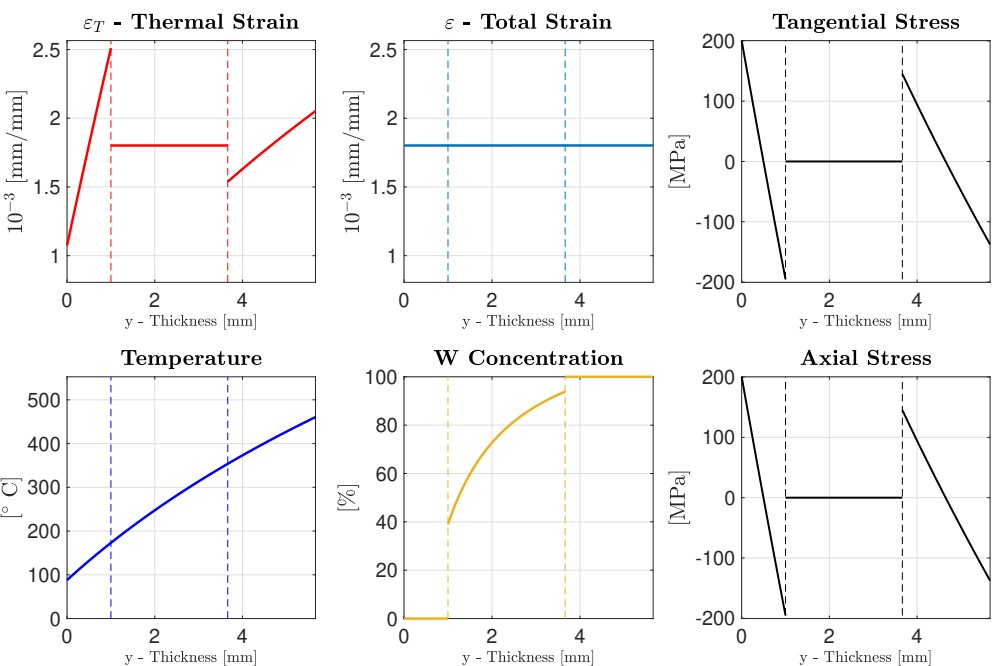

**Figure 2.** Results for a thin cylindrical body with a homogeneous thermal load. The armor is made of tungsten and the heat sink is made of AISI316L. Inputs: $\delta_1 = 2$ mm, $\delta_2 = 1$ mm; $q = 1.4$ MWm$^{-2}$; $a = 1.502 \times 10^{-2}$ Wm$^{-1}$K$^{-2}$; $b = 13.98$ Wm$^{-1}$K$^{-1}$; $T_{water} = 60$ °C; $h = 50$ kWm$^{-2}$K$^{-1}$; $R = 15.25$ mm; $T_0 = 20$ °C. The temperature field is the one indicated in Equation (12). The material properties of AISI316L and tungsten are taken from [6]. The FGM is made of a W/AISI316L mixture. The interlayer thickness results to be $\delta_3 = 2.66$ mm. The geometrical boundaries between the sub-components (heat sink/FGM/armor) are indicated by dashed lines.

$$T(y) = \frac{b + \sqrt{b^2 - 2a\left[c + qR\ln\left(1 + \frac{y}{R}\right)\right]}}{a} \; ; \; c = -\frac{a}{2}\left(\frac{q}{h} + T_{water}\right)^2 - b\left(\frac{q}{h} + T_{water}\right) \quad (12)$$

which is the analytical solution of the steady state heat equation for a homogeneously loaded hollow cylinder with inner radius $R$ and having a thermal conductivity $\lambda(T) = aT + b$. The coefficients $a$ and $b$ are chosen to be the one of AISI316L [6]. $q$ is the heat flux absorbed at the wall interface with water, $T_{water}$ is the water bulk temperature and $h$ is the heat transfer coefficient due to convection. Figure 2 shows that the design rationale of having a constant total strain over the cylinder thickness is satisfied. The ideal W concentration is computed in such a way that the FGM is stress-free, since it is derived by keeping $\varepsilon_T = \varepsilon_0$ in the whole interlayer. One observes that $C(y)$, computed using Equation (11), does not have as boundary conditions $C = 0$ at the interface with the heat sink or $C = 1$ next to the armor. In previous FGM design studies [10,20] these conditions were applied to the concentration function, but we find this to be deleterious for the thermal stresses. In fact, to match the total strain we showed that the mean thermal strain of the bodies must be matched, and no need for the continuity of the thermal strain field is required even in an ideal case. The material distribution was also found discontinuous in [11], and here a thermomechanical justification of this effect is found. Naturally, the fabrication of a FGM having a continuous concentration function in space is not technically feasible. Therefore, the ideal concentration function must be somehow discretized. This unavoidably would lead to the generation of thermal stresses in the FGM layers due to the thermal gradient, while keeping equal to zero the contribution generated by the CTE mismatch. The more layers are chosen for a better approximation of the ideal concentration, the less will be the stresses inside the interlayer. However, as long as one keeps

the total strain constant also during this discretization, the stresses in the armor and the heat sink will not be affected. The number of layers is therefore only a function of the maximum desired stress inside the FGM.

### 2.2. Thin Plate Body with a One-Sided Thermal Load

Another relevant case for the PFCs can be found in a plate having a monotonic temperature field over its thickness. A one-sided thermal boundary condition corresponds to the real situation which occurs in a fusion reactor, since the PFCs are loaded only on the surface facing the plasma. This results in a hotter and a colder side of the component, which in turn leads to the bending of the module. Therefore, even if one considers the thickness $\delta$ to be much smaller than the other dimensions of the plate, the total strain in the principal directions will not be constant anymore over $y$. The constitutive equations in Cartesian coordinates $(x,y,z)$, for a thin plate are obtained by considering the normal stress in the thickness direction to be negligible compared the other principal stresses ($\sigma_y \sim 0$):

$$\begin{cases} \varepsilon_x &= \frac{1}{E}(\sigma_x - \nu\sigma_z) + \alpha(T - T_0) \\ \varepsilon_z &= \frac{1}{E}(\sigma_z - \nu\sigma_x) + \alpha(T - T_0) \end{cases} \tag{13}$$

According to the plate theory [21], the normal strain in the $x$ and $z$ direction have the following form:

$$\begin{cases} \varepsilon_x = \varepsilon_{0,x} + \frac{y}{r_x} \\ \varepsilon_z = \varepsilon_{0,z} + \frac{y}{r_z} \end{cases} \tag{14}$$

where $\varepsilon_{0,x}$ and $\varepsilon_{0,z}$ are the normal strains at $y = 0$ in the $x$ and $z$ directions, $r_x$ is the radius of curvature of the plate in a plane parallel to the $xy$ plane and $r_z$ is the radius of curvature in a plane parallel to the $zy$ plane. Analogously to the previous case, the temperature is a function of $y$ only, and therefore $\varepsilon_x$ and $\varepsilon_z$ do not depend on $x$ or $z$. Moreover, the $r_i$ can be considered constant over the thickness, being valid the hypothesis of a thin plate. In absence of external mechanical loads, both the integral of the principal stresses over the plate thickness and the bending moments are zero in the whole body. These equilibrium conditions, together with Equation (13), allow to obtain the unknown variables in Equation (14):

$$\begin{cases} \varepsilon_0 = \varepsilon_{0,x} = \varepsilon_{0,z} = \frac{1}{\delta} \int_{-\frac{\delta}{2}}^{\frac{\delta}{2}} \alpha(T - T_0)dy \\ \frac{1}{r} = \frac{1}{r_x} = \frac{1}{r_z} = \frac{12}{\delta^3} \int_{-\frac{\delta}{2}}^{\frac{\delta}{2}} \alpha(T - T_0)ydy \end{cases} \tag{15}$$

For a simpler representation of the integrals in Equation (15) the frame of reference is chosen with $y = 0$ at the mean plane of the plate. In an analogous fashion as the previous case, one notices that $\varepsilon_0$ corresponds to the mean thermal strain of the body $\varepsilon_{T,m}$. Additionally, one observes that the total strains in the $x$ and $z$ directions result to be equal to each other, and therefore the plate is bent to a spherical shell. This result is valid for the armor and the heat sink, since both are considered thin plates. If they were not bonded to each other, each sub-component would deform freely in the described fashion, even if having a different strain of the mean plane $\varepsilon_0$ and curvature $\frac{1}{r}$. In order to eliminate the stresses due to the CTE difference, one must engineer the component such that the total strain field of the PFM and the heat sink are matched. This would require the curvature of the sub-components to be equal, together with a condition on the mean thermal strains of the bodies:

$$\begin{cases} \frac{1}{r_1} = \frac{1}{r_2} = \frac{1}{r} \\ \varepsilon_{T,m1} = \varepsilon_{T,m2} + \frac{\Delta y}{r} \end{cases} \tag{16}$$

where the subscript 1 refers to the armor and 2 to the heat sink material, and $\Delta y$ is the thickness between the location in the heat sink where $\varepsilon_T = \varepsilon_{T,m2}$ and the point in the PFM

where $\varepsilon_T = \varepsilon_{T,m1}$. We have extensively discussed in the previous case how it is possible to modify the W mean thermal strain $\varepsilon_{T,m1}$ by engineering the thickness of an interlayer interposed between the sub-components, and the same approach can be used for this instance. However, the equation concerning the radii of curvature cannot be generally satisfied. Let us semplify the description by considering the materials to have a constant thermal conductivity $\lambda$ and CTE. In this linear approach, the temperature field is:

$$T(y) \sim \frac{q}{\lambda}y + T_m \tag{17}$$

where $q$ is the impinging heat flux and $T_m$ the mean temperature of the plate. Using Equation (15), one obtains:

$$\begin{cases} \varepsilon_0 = \varepsilon_{T,m} = \alpha(T_m - T_0) \\ \dfrac{1}{r} = \dfrac{\alpha}{\lambda}q \end{cases} \tag{18}$$

The curvature depends only on the applied power density $q$, fixed by the plasma, and on the material properties, namely the ratio $\dfrac{\alpha}{\lambda}$ between the CTE and the thermal conductivity. Neither the geometry nor the temperature field can be used to tailor the bending of the plate. When the total strain of the "free-to-deform" bodies has a curvature term, an interlayer capable of bringing to zero the thermal stresses due to the CTE mismatch can be developed only if the PFM and heat sink material are such that:

$$\frac{\alpha}{\lambda}\bigg|_1 \sim \frac{\alpha}{\lambda}\bigg|_2 \tag{19}$$

Such a relation is not generally satisfied. Table 1 reports the values of $\frac{\alpha}{\lambda}$ for several fusion-relevant materials. Unfortunately, it is possible to assess that there is no direct matching. Therefore, the total strain field of the armor and heat sink cannot be paired, at least generally, as was performed in the previous case. The design rationale that we propose to tackle this issue consists in constraining the PFCs in such a way that bending is not allowed. This is achievable, for example, by external kinematic conditions. It will then be necessary that a proper fixation system is designed to best achieve this condition in a real component in the reactor. Such an approach would result in the increase of the contribution to the thermal stresses due to the gradient, both in the armor and in the heat sink, because the free bending of the plate would be hindered. The benefit however lies in the possibility of reducing to zero the stresses due to the mismatch of the CTE. In such a way, the critical interface of joining between the PFM and the rest of the PFC, generally prone to detachments, does not experience any intensification of the thermomechanical stresses. With the absence of a curvature, the case of the plate degenerates into the previous one. The design rationale again becomes the matching, among the sub-components, of the total strain $\varepsilon_0$. As a first step, the thermomechanical design of the heat sink is carried out since it does not depend on the other sub-components, but only on $T(y)$ which in our description is considered an input. Therefore, the geometry of the heat sink is determined by satisfying the design criteria while the temperature field is applied and no bending is imposed. The mean thermal strain of the heat sink $\varepsilon_{T,m2}$ is chosen as the target value of the total strain $\varepsilon_0$, which is kept constant in the whole PFC. The interlayer, in its thickness and composition, is designed by using the Equations (9) and (11) derived previously. In the same fashion as before, the ideal W concentration allows for the theoretical determination of a stress-free FGM. By hindering the bending of the plate, the engineering of a FGM interlayer able to eliminate the stresses due to the CTE mismatch is then possible.

**Table 1.** Values of $\frac{\alpha}{\lambda}$ for several fusion-relevant materials. Material data from [6,22].

| $\frac{\alpha}{\lambda}$ [$10^{-2}$ µmW$^{-1}$] | W | Cu | CuCrZr | AISI316L | EUROFER97 |
|---|---|---|---|---|---|
| T = 20 °C | 2.60 | 4.16 | 5.25 | 107.14 | 36.53 |
| T = 200 °C | 2.90 | 4.56 | 5.16 | 97.76 | 35.86 |

## 3. Conceptual Design of a Steel-Based PFC with a Engineered W/Steel FGM Interlayer

In this section, we apply the methodology just described to a more realistic conceptual design of a PFC having a steel plate with grooved coolant channels as heat sink. The main differences from the simple ideal case lie in the fact that now the body will be multiply connected and the temperature field $T$ will not depend only on the plate thickness. These hypothesis would result in a total strain tensor that has not anymore equal components along the $x$ and $z$ directions. Another issue that will be solved is the discretization of the ideal FGM in a multi-layer FGM having discrete concentrations.

The chosen layout for the PFC is the flat-tile configuration, with the heat sink being a steel plate with squared conduits for the water. The geometry of the component is sketched in Figure 3. The choice of the squared shape of the coolant channel is to lower the spread between the water bulk temperature, which is a good approximation of the mean temperature of the heat sink, and the maximum temperature of the steel domain. In such a way, one minimizes the thermal stresses due to the gradient since they depend on the difference between the local temperature and the average temperature of the body $\sim (T - T_m)$, as we discussed previously.

The materials used for the design are AISI316L for the heat sink and Plasma Sprayed tungsten for the armor. This austenitic steel, which has a very low thermal conductivity, is characterized by low costs and high availability. Moreover, the choice of this material for the heat sink is also for testing the robustness of the proposed design methodology and assessing its capability of designing a PFC with such a thermally insulating material ($\lambda \sim 15$ Wm$^{-1}$K$^{-1}$). Plasma Spraying is a coating technique able to deposit millimeters-thick coatings with high density [23]. The material properties of AISI316L and of W will be taken from [6]. The PFM properties are chosen to be the one of bulk W, with the exception of the thermal conductivity, which must be modified to reproduce the performance of the plasma sprayed coating. In fact, it has been observed experimentally that the conductivity of plasma sprayed W is far below the one of the hot-rolled material. In particular, measurements show [8,10] that this thermophysical property is, for the coating, very similar to the one of steel ($\lambda \sim 15$ Wm$^{-1}$K$^{-1}$). Therefore, due to the lack of a specific database, one will consider the thermal conductivity of the armor to be equal to that of the heat sink. Such a hypothesis is also useful to decouple the thermal problem from the thermomechanical one. In this way, the temperature field will not depend on the FGM concentration function. As mentioned before, the coupled problem can be easily solved with an iterative algorithm, converging on both the FGM thickness and concentration. However the implementation of such numerical scheme is outside the scope of this paper.

The design inputs we chose for this study are the following:

- Heat load coming from the plasma $q = 0.5$ MWm$^{-2}$
- Operative coolant pressure $p = 4$ MPa
- Operative water bulk temperature $T_{water} = 60$ °C
- Heat transfer coefficient of convection $h = 50$ kWm$^{-2}$K$^{-1}$
- Stress free temperature $T_0 = 20$ °C
- Thickness of the tungsten armor $\delta_1 = 2$ mm

Which correspond to possible boundary conditions for a first wall module where the impinging load is mainly due to radiation and little to no charged particle irradiation is present. The heat load $q$ is considered constant on the whole plasma-facing surface.

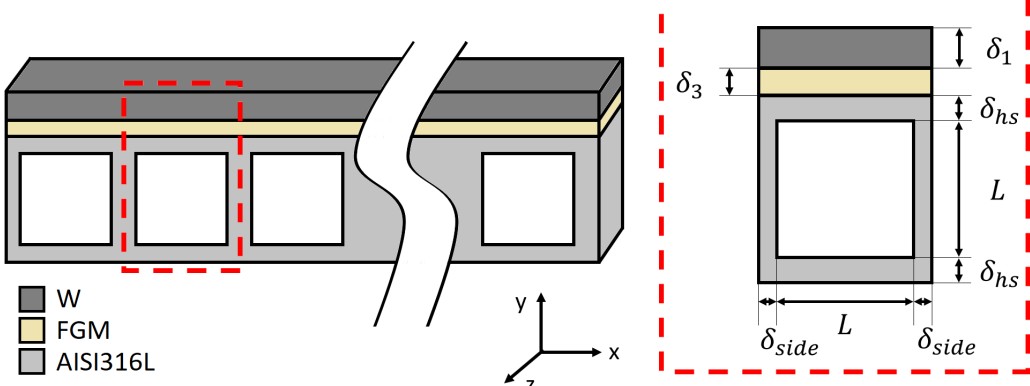

**Figure 3.** Sketch of the geometry of the PFC, together with a magnification of the elementary cell. The nomenclature of all the dimensional parameters is also shown.

### 3.1. Design of the Heat Sink

Under the above mentioned assumptions, the geometry of the heat sink is independent from the interlayer, whose thickness $\delta_3$ and W concentration $C$ are yet to be defined. The design of the heat sink is carried out by satisfying the design criteria even if no interlayer and armor were joined to it. The heat sink is the structural material in these components and has to ensure the capability to manage the primary and secondary stresses even if both the PFM and FGM were to fail and detach from it. The main parameters of Figure 3 that drive the stresses in the steel plate are the thickness $\delta_{hs}$ of the upper/lower part of the heat sink and the ratio $\left(\frac{L}{\delta_{hs}}\right)$. The first variable impacts the secondary stresses, while the latter mainly defines the primary loads which are the one due to the pressure applied by the coolant. The determination of $\delta_{hs}$ and $\left(\frac{L}{\delta_{hs}}\right)$ is carried out in such a way that the design criteria [6] are satisfied. The process is explained in details in [24] and will not be reported here for brevity. Using the primary membrane plus bending stress criterion along with the membrane primary plus secondary membrane stress criterion one obtains 1 mm for the heat sink thickness and 7 for $\left(\frac{L}{\delta_{hs}}\right)$. One must point out that for the allowable stress intensity $S_e$ we used conservatively the definition of one third of the minimum ultimate tensile strength, to take into account a high neutron exposure which could lead to a minimum uniform elongation of less than two percent. For the complete geometry of the steel sub-component, only the parameter $\delta_{side}$ is left to be determined. The impact of this quantity is mainly on the temperature field. The smaller $\delta_{side}$ is, the lower would be the maximum temperature of the heat sink, and also the temperature field would depend more weakly on $x$, $T \sim T(y)$. In this work we chose arbitrarily $\delta_{side} = 2\delta_{hs}$. To check if all criteria are indeed satisfied, we then proceed by importing the computed geometry in a FEM thermomechanical model of the steel heat sink. The numerical model is carried out in ANSYS Workbench 2022 R1, using only an elementary cell which has been extruded along the $z$ direction for a length $L$, chosen arbitrarily equal to the channel width. The kinematic boundary conditions used for the thermomechanical model are the following:

- No bending: the $y$ component of the displacement is zero on the backside of the plate, which are the points $(x, y = 0, z)$.
- Symmetry condition: the $x$ component of the displacement is zero on the points $(x = 0, y, z)$.
- Symmetry condition: the $z$ component of the displacement is zero on the points $(x, y, z = 0)$.
- To simulate an arbitrary repetition of the elementary cell both along the $x$ and the $z$ direction:
  - $z$ direction: the $z$ component of the displacement is constant on the points $(x, y, z = L)$.

- *x* direction: the *x* component of the displacement is constant on the points $(x = L + 2\delta_{side}, y, z)$.

By applying the last two conditions, we are admittedly ignoring the boundary effects that can occur at the edges of a plate, where points of the body that were on a line perpendicular to the middle surface of the plate do not stay in such a configuration after the application of the temperature field. These effects depend on the real geometry of the whole plate and would be computed at a later stage of the design than the conceptual phase. The mesh of the model is shown in Figure 4, along with the temperature field $T(x, y)$ inside the steel plate. Assuming a negligible heat peaking factor in the solid domain of the interlayer and the armor, which is valid for poor thermal conductors, the load has been applied directly to the heat sink. The mesh is made of 11200 SOLID186 elements, corresponding to more than 80 k nodes. The results of the thermomechanical problem are shown in Table 2, where the linearized stress in the most critical sections are reported. The paths used for the linearization can be found in Figure 5. In Table 3 one can assess that the design criteria are satisfied, since the ratio between the linearized stresses and the allowable stresses is less than one for all the paths. For the steel heat sink, the secondary stresses result to be the major contribution to the thermomechanical loads, due to the rather low thermal conductivity of the material. Specifically, the primary plus secondary membrane stress criterion is the most stringent one for the component, having the lowest margin.

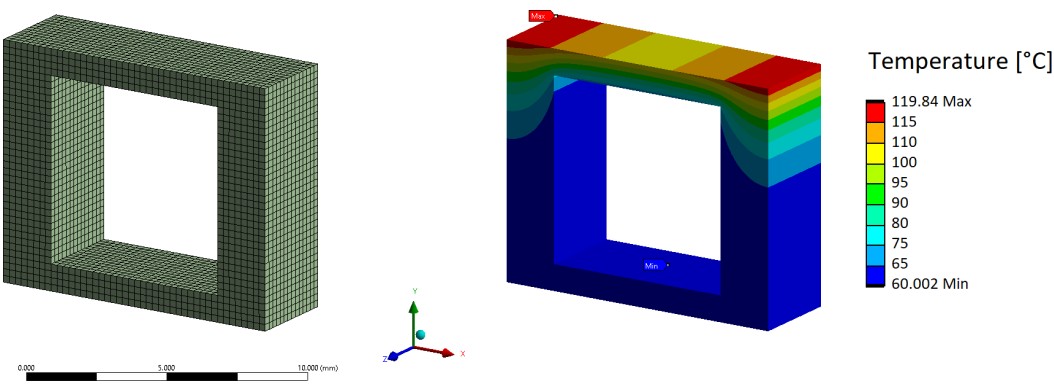

**Figure 4.** (**Left**) Mesh used for the FEM model of the steel heat sink, without both interlayer and armor.( **Right**) Temperature field inside the heat sink.

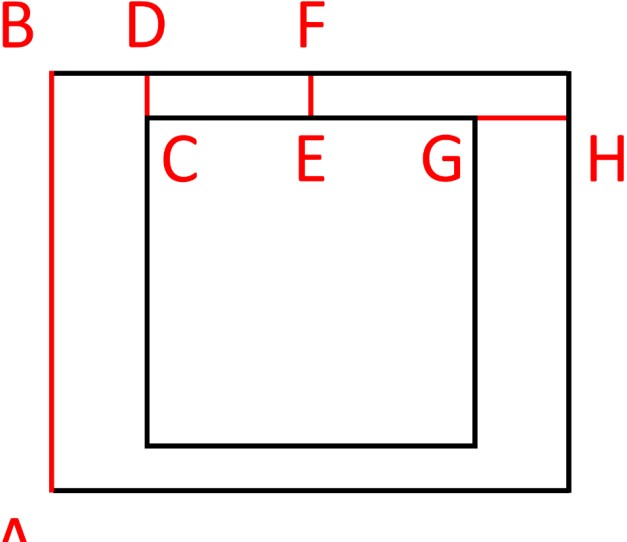

**Figure 5.** Supporting line segments chosen for the stress linearization.

**Table 2.** Linearized stress in the steel heat sink without armor and interlayer. Paths are indicated in Figure 5.

| | No Interlayer and Armor | | |
|---|---|---|---|
| Path | $\overline{P_m + Q_m}$ [MPa] | $\overline{P_m + P_b}$ [MPa] | $\overline{Q}$ [MPa] |
| $\overline{AB}$ | 0.4259 | 7.21 | 72.034 |
| $\overline{CD}$ | 119.36 | 63.932 | 169.25 |
| $\overline{EF}$ | 68.823 | 70.384 | 165.9 |
| $\overline{GH}$ | 60.254 | 53.752 | 80.898 |

**Table 3.** Verification of the design criteria on the most critical sections. $T_{avg}$ is the average temperature along the path; $S_m$ is the allowable membrane stress; $S_e$ is the allowable stress intensity. Conservatively $K_{eff}$ is taken as 1. Paths are indicated in Figure 5.

| Path | $T_{avg}$ [°C] | $S_m$ [MPa] | $S_e$ [MPa] | $\dfrac{\overline{P_m + Q_m}}{S_e}$ | $\dfrac{\overline{P_m + P_b}}{K_{eff}S_m}$ | $\dfrac{\overline{P_m + P_b + \overline{Q}}}{3S_m}$ |
|---|---|---|---|---|---|---|
| $\overline{AB}$ | 72.034 | 127 | 146.05 | 0.003 | 0.0568 | 0.2080 |
| $\overline{CD}$ | 95.367 | 127 | 138.50 | 0.8618 | 0.5034 | 0.6120 |
| $\overline{EF}$ | 88.458 | 127 | 140.75 | 0.4890 | 0.5542 | 0.6202 |
| $\overline{GH}$ | 90.707 | 127 | 140.02 | 0.4303 | 0.4232 | 0.3534 |

*3.2. Engineering of the FGM Interlayer*

Once the steel heat sink has been sized, one proceeds in designing the interlayer. This part will be engineered so that the whole component would have a total strain, due to the secondary loads, as much as possible similar to the one of the heat sink without any armor. Due to the boundary conditions, thermal and kinematic, that we applied, the total strain of the steel sub-component in the $z$ direction is constant on the whole plate. However, due to the dependence on both $x$ and $y$ of the temperature field, as shown in Figure 4, the total strain is not homogeneous in such directions. Figure 6 shows the principal total strains of the steel body, without any interlayer or armor, only due to the effect of the temperature field. One observes that the total strain along z is different from the mean total strain in the $x$ direction. This means that the expansion of the heat sink is anisotropic. The rectangular surface of the steel plate stays rectangular after the application of the temperature field, but the ratio between homologous sides is not the same. Coupling an isotropic material, the interlayer, to such total strain field will unavoidably lead to additional stresses even in an ideal case. Apart from this effect, which cannot be mitigated, one engineers an interlayer that would not add any additional loads to the heat sink. This is done by designing the component in such a way that the total strain of both the armor and heat sink would be similar to the one of the steel body without the PFM. Therefore, as explained before, we impose that the mean thermal strain of the FGM and of the W to be equal of a target value $\varepsilon_0$. In this study, this target parameter is chosen equal to the total strain $\varepsilon_x$ of the steel body, averaged inside an elementary cell:

$$\varepsilon_0 = \frac{1}{L + 2\delta_{side}} \int_0^{L+2\delta_{side}} \varepsilon_x dx \qquad (20)$$

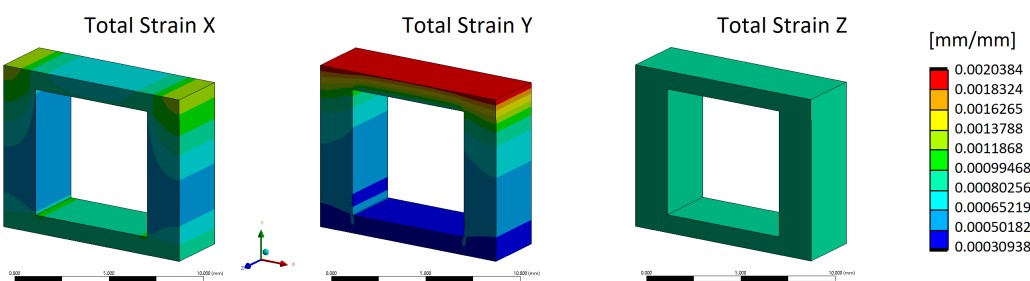

**Figure 6.** Principal components of the total strain tensor of the steel heat sink, when the temperature field is applied without any interlayer and armor.

Our choice results in the total deformation of the interlayer matching, on average, the one of the heat sink along $x$. However, since this value is higher than $\varepsilon_z$, some stresses along $z$ due to the mismatch of the thermal expansion will arise. The heat sink will be tractioned by the interlayer and an analogous compression would occur inside the FGM. However, such configuration is beneficial, at least for the heat sink, because, due to the prevented bending by the external kinematic constraints, the steel sub-component is axially compressed under the thermal gardient. Stresses normal to $z$ in the plate will than lower at the interface, with a compressed FGM being the price to be paid. Equation (20) provides a value of the target strain of $9.207 \times 10^{-4}$ mm/mm. The interlayer thickness $\delta_3$ is obtained in the same fashion of the thin hollow cylinder. Therefore one solves:

$$\varepsilon_{T,m1} = \varepsilon_0$$

$$\frac{1}{\delta_1(L + 2\delta_{side})} \int_{\delta_2+\delta_3}^{\delta_1+\delta_2+\delta_3} \int_0^{L+2\delta_{side}} \alpha_1(T)(T - T_0)dxdy = \varepsilon_0 \tag{21}$$

where the thickness of the whole heat sink $\delta_2$ is in this case equal to $L + 2\delta_{hs}$. The temperature field is computed using a thermal FEM model. In such computations, the interlayer thickness is increased gradually up to the point that the mean thermal strain of the PFM is equal to $\varepsilon_0$. The requirement is met for $\delta_3 \sim 2.68$ mm, corresponding to a W mean temperature of 222.74 °C. For our design the interlayer thickness is rounded to 2.7 mm. The whole geometry of the component is now completely defined. Subsequently, one proceeds in determining the concentration of the FGM. To do so, one consistently uses the condition of the mean thermal strain to be equal to $\varepsilon_0$. However, as explained previously, a stronger condition can be put for the composite interlayer. For an ideal FGM, varying continuously its concentration in space, the function $C$ can be tailored such that the thermal strain of this material would be constant and equal to the target value. Automatically, the constraint on the mean value would be satisfied. If the chosen manufacturing technique for the coating is the plasma spraying, the concentration of the FGM can vary only with the thickness $C = C(y)$. Such a function is derived by solving Equation (11). For this purpose, a 1D approximation is required of the temperature field in the solid sub-domain having as boundaries the steel/water interface and the plasma-facing surface. We use:

$$T(y) = \frac{b + \sqrt{b^2 - 2a(qy + c)}}{a} \; ; \; c = -\frac{a}{2}T_{wall}^2 - bT_{wall} \tag{22}$$

which is the analytical solution of the steady state heat equation of a plate loaded from one side and having a thermal conductivity $\lambda(T) = aT + b$. The coefficients $a$ and $b$ are chosen to be the one of AISI316L [6]. $q$ is the heat flux impinging on the plasma-facing surface and $T_{wall}$ is the mean temperature at the interface between water and steel and the frame of reference is chosen such that $y = 0$ at this location. The resulting concentration profile is shown in Figure 7, together with the corresponding thermal strain that in this ideal case is constant in space and equal to $\varepsilon_0$. However, a material with such continuously varying composition is very challenging to manufacture. Therefore, the function $C(y)$ must be discretized in space.

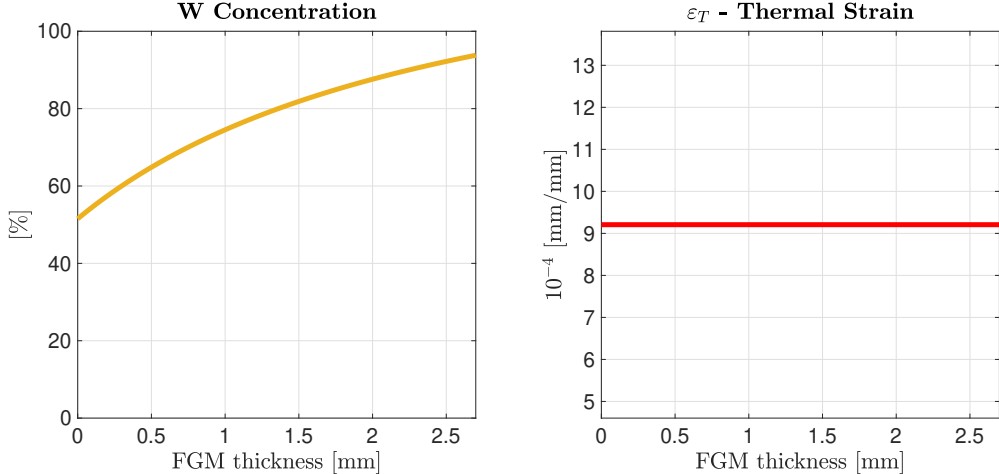

**Figure 7.** Ideal W concentration function of the FGM derived using Equation (11). The temperature field is derived from Equation (22), with: $T_{wall} = 111.56\,°C$; $q = 0.5\,MWm^{-2}$; $a = 1.502 \times 10^{-2}\,Wm^{-1}K^{-2}$; $b = 13.98\,Wm^{-1}K^{-1}$; $T_0 = 20\,°C$. $T_{wall}$ is the computed average temperature at the interface steel/water in the FEM model (Figure 4). The material properties of AISI316L and tungsten are taken from [6]. The FGM is made of a W/AISI316L mixture. The interlayer thickness results to be $\delta_3 = 2.7\,mm$.

### 3.3. Discretization of the Ideal FGM in Layers

The simplest discretization would consist in a FGM monolayer. For such a material, the thermal strain is equal to:

$$\varepsilon_{T,3}(T) = \alpha(T)(T - T_0) = [(\alpha_1(T) - \alpha_2(T))\,C_1 + \alpha_2(T)](T - T_0) \tag{23}$$

As determined by our design methodology, the mean thermal strain of the interlayer $\varepsilon_{T,m3}$ must be equal to the target strain. One therefore obtains:

$$\frac{1}{\delta_3} \int_{\delta_2}^{\delta_2+\delta_3} \varepsilon_{T,3}(T)dy = \varepsilon_0$$

$$C_1 = \frac{\delta_3 \varepsilon_0 - \int_{\delta_2}^{\delta_2+\delta_3} \alpha_2(T)(T - T_0)dy}{\int_{\delta_2}^{\delta_2+\delta_3} [\alpha_1(T) - \alpha_2(T)](T - T_0)dy} \tag{24}$$

If more than one FGM layer is desired, the process is analogous. For each $i$-th layer the condition on the mean value of the thermal strain must be satisfied. The equations to determine the layer concentration $C_i$ in the case of $N$ total layers are therefore the following:

$$\frac{1}{y_i - y_{i-1}} \int_{y_{i-1}}^{y_i} \varepsilon_{T,i}(T)dy = \varepsilon_0 \rightarrow \begin{cases} C_1 &= \dfrac{(y_1 - \delta_2)\varepsilon_0 - \int_{\delta_2}^{y_1} \alpha_2(T)(T - T_0)dy}{\int_{\delta_2}^{y_1} [\alpha_1(T) - \alpha_2(T)](T - T_0)dy} \\[4mm] C_i &= \dfrac{(y_i - y_{i-1})\varepsilon_0 - \int_{y_{i-1}}^{y_i} \alpha_2(T)(T - T_0)dy}{\int_{y_{i-1}}^{y_i} [\alpha_1(T) - \alpha_2(T)](T - T_0)dy} \\[4mm] C_N &= \dfrac{(\delta_2 + \delta_3 - y_{N-1})\varepsilon_0 - \int_{y_{N-1}}^{\delta_2+\delta_3} \alpha_2(T)(T - T_0)dy}{\int_{y_{N-1}}^{\delta_2+\delta_3} [\alpha_1(T) - \alpha_2(T)](T - T_0)dy} \end{cases} \tag{25}$$

where $y_i$ is the spacial coordinate at which the $i$-th layer ends. Moreover, it is appropriate to size the different layers in a way that the stresses are distributed evenly. Therefore the choice of the $y_i$ is not arbitrary. Since the stresses depend on the difference between the total and the thermal strain, this constraint corresponds to the condition of equal absolute maximum of the thermal strain of the different FGMs:

$$\max[\varepsilon_{T,i}(T)] = \max[\varepsilon_{T,i+1}(T)]$$
$$\alpha_i(\,T(y_i)\,)[T(y_i) - T_0] = \alpha_{i+1}(\,T(y_{i+1})\,)[T(y_{i+1}) - T_0] \tag{26}$$

where $\alpha_i$ is the secant CTE of the $i$-th layer. Equations (25) and (26) allow to find all the $C_i$ and $y_i$ to size the interlayer and can be solved iteratively. Figure 8 shows the discretization of $C(y)$ up to four layers.

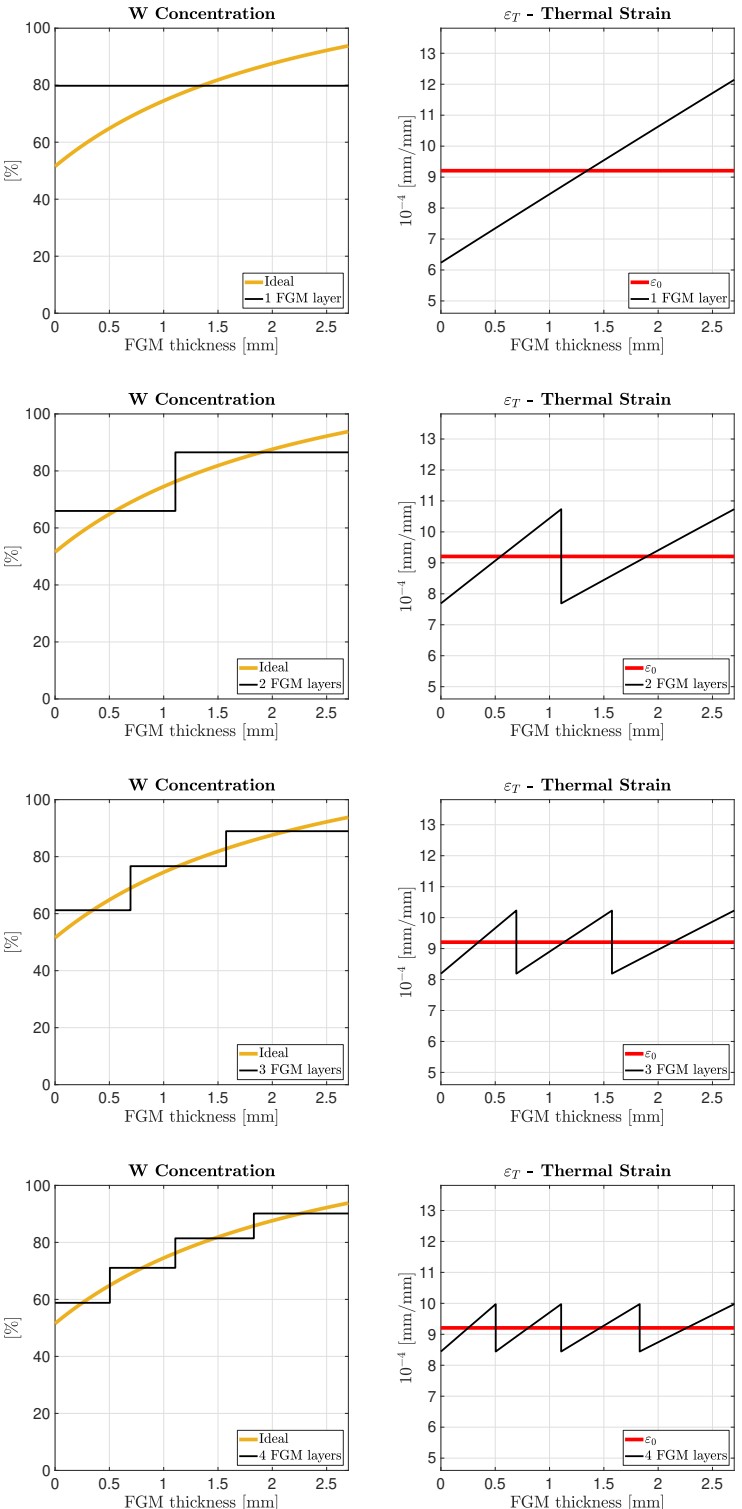

**Figure 8.** Discretization of the ideal concentration of the FGM in a number of layers, following Equations (25) and (26).

It should be highlighted that the amount of layers would not impact substantially the stresses inside the steel heat sink and the tungsten armor. This is because, by construction, each layer will have the same exact total strain, whether one chooses a monolayer or more. The difference will be in the stresses due to the thermal gradient inside the FGM, which depend on the difference between the thermal strain and $\varepsilon_0$. The higher is the number of layers, the closer the FGM will be to the ideal condition where $\varepsilon_{T,3}(T) = \text{cost} = \varepsilon_0$. One also notices that due to the condition of equal stress among the FGMs, the layers with a higher steel content are thinner than the others. The explanation of this is in the fact that these FGMs would increase their thermal strain "faster", due to a higher CTE, and therefore meet earlier in space the maximum $\varepsilon_T$ which is kept the same for all FGMs.

### 3.4. Mechanical Results

The FEM model used for the thermomechanical simulation uses the same thermal boundary conditions described at the beginning of this section. The kinematic constraints applied to the whole PFC, equipped now with both interlayer and armor, coincide with the ones applied for the simulation with just the steel heat sink. Figure 9 shows the mesh and the resulting temperature field in the PFC. The mesh is made of 22904 SOLID186 elements, accounting for more than 100 k nodes. The Young's Modulus of the plasma sprayed W and of all FGMs is assumed to be equal to that of the hot rolled tungsten (available in [6]). Admittedly, this is a conservative hypothesis to test the robustness of the design method we are proposing. Hot-rolled tungsten has in fact an *E* which is almost double the Young's modulus of steel. Therefore, being the thermal stresses proportional to the rigidity of the bodies, as one can ascertain from Equation (7), we are consciously overestimating such loads.

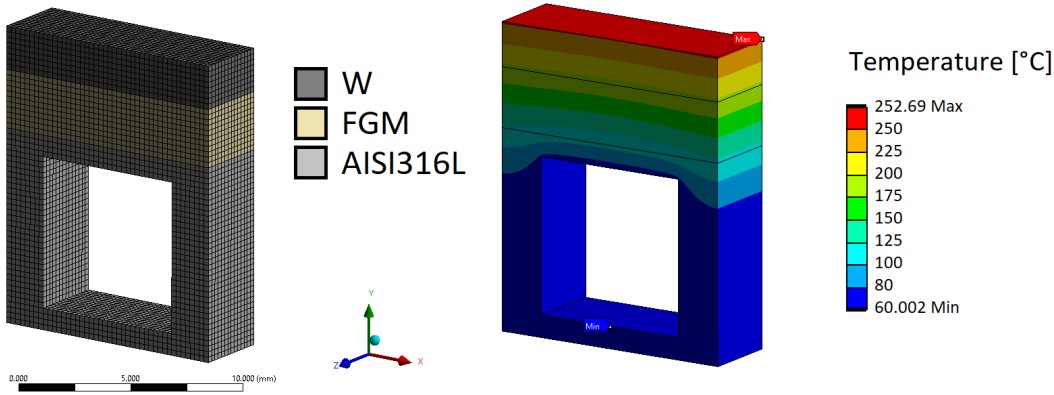

**Figure 9.** (**Left**) Mesh used for the FEM model of the whole PFC, having a FGM interlayer (yellow) and a W armor (dark grey). (**Right**) Temperature field in the whole component.

A first computation is carried out in the case of an ideal FGM. To do so, we impose numerically to the interlayer a thermal strain equal to $\varepsilon_0$. On the heat sink and the armor, the thermal field is imported in order to compute the real $\varepsilon_T(T)$. The linearized stresses computed at the different support line segments are reported in Table 4. Comparing Table 2 to Table 4, one notices that the linearized secondary stresses do not change significantly between the cases, especially on the most loaded section $\overline{CD}$, meaning that the addition of both the armor and the interlayer has not intesified the thermal stresses. Naturally, the primary stresses experience a substantial decrease due to a thicker resistant section which balances the external forces. For completeness, the contour of the von Mises stress on the whole component is shown in Figure 10. One notices that even in this ideal case, the FGM is not stress free. As explained previously, this is due to both the anisotropy in the total strain of the "free-to-deform" heat sink ($\varepsilon_x \neq \varepsilon_z$) and the manufacturing constraint of having the concentration function only dependent on $y$, and not on $(x, y)$.

**Table 4.** Linearized stress in the steel heat sink with an ideal FGM. Paths are indicated in Figure 5.

| | Ideal FGM | | |
|---|---|---|---|
| Path | $\overline{P_m + Q_m}$ [MPa] | $\overline{P_m + P_b}$ [MPa] | $\overline{Q}$ [MPa] |
| $\overline{AB}$ | 15.891 | 10.362 | 86.761 |
| $\overline{CD}$ | 117.50 | 14.510 | 169.77 |
| $\overline{EF}$ | 53.460 | 1.4331 | 133.26 |
| $\overline{GH}$ | 47.635 | 9.6404 | 52.553 |

**Figure 10.** Contour of the von Mises stress in the component due to the heat load and the pressure. Case with an ideal FGM.

The discretization of the concentration function has been carried out using Equations (25) and (26). Up to four FGM layers have been evaluated. Table 5 reports the results of the discretization process, which are the data represented graphically in Figure 8. For the four cases, the linearized stresses in the steel heat sink are shown in Table 6. One notices that, as expected by the design methodology, the steel heat sink is virtually unaffected by the several discretizations. This is a direct consequence of the engineering of the interlayer, which was designed to have the same total strain due to the temperature field, regardless of the number of layers. Table 6 verifies that keeping the mean thermal strain constant among the different layers, indeed ensures that the stresses in the heat sink are only the ones due to the thermal gradient. One can also compare such results with the ideal FGM case (Table 4), finding again marginal variations. All the design criteria for the structural steel sub-component are therefore satisfied even in these four cases. Furthermore, the stresses inside the armor are virtually independent from the discretization. A substantial difference depending on the amount of layers is instead found in the loads inside the FGM. As mentioned before, the stresses due to the thermal gradient increase with the maximum thermal strain of each single layer. Such parameter is reduced with an increasing discretization of the concentration function, reducing the loads in the FGM. Figure 11 shows the contours of the von Mises stress on the whole PFC, for different amounts of FGM layers. One observes that only the loads inside the interlayer are impacted by the different number of layers, while the contour inside the armor and heat sink is virtually unmodified. Therefore, during the design phase, the amount of divisions in the FGM is determined only by the desired stress inside the interlayer, with the minimum limit being the ideal FGM case.

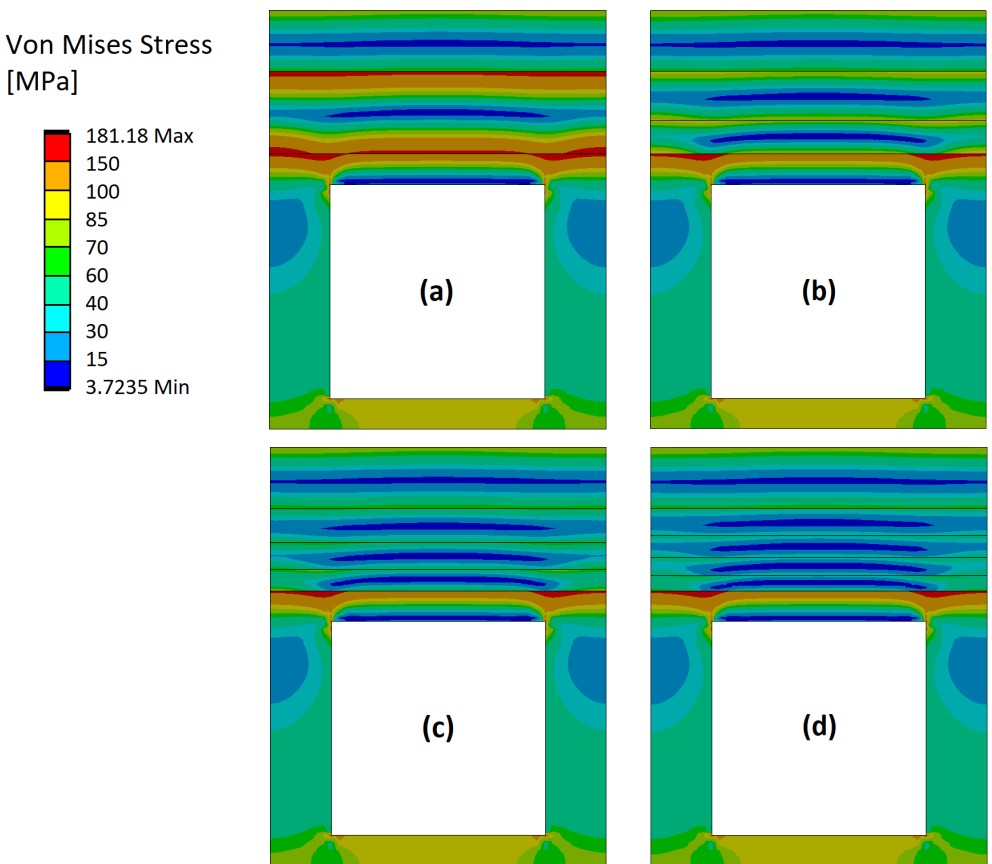

**Figure 11.** Contour of the von Mises stress in the component due to the heat load and the pressure. Case with discretized FGMs: (**a**) One FGM layer. (**b**) Two FGM layers. (**c**) Three FGM layers. (**d**) Four FGM layers. One notices that the von Mises stresses both in the armor and the heat sink are virtually constant for all the different discretizations.

**Table 5.** Discretization of the FGM interlayer up to four sub-layers (same data are also visually represented in Figure 8). For each layer, the concentration $C_i$ as well as the thickness $\Delta t_i$, rounded from the value derived trough the computation, are reported.

| | $C_i$ | $y_i$ [mm] | $y_i$ (Rounded) [mm] | $\Delta t_i$ [mm] |
|---|---|---|---|---|
| **1 FGM layer** | $C_1 = 79.798\%$ | - | - | $\Delta t_1 = 2.7$ |
| **2 FGM layers** | $C_1 = 65.958\%$ <br> $C_2 = 86.523\%$ | $y_1 = 1.107$ <br> $y_2 = 2.7$ | $y_1 = 1.1$ <br> $y_2 = 2.7$ | $\Delta t_1 = 1.1$ <br> $\Delta t_2 = 1.6$ |
| **3 FGM layers** | $C_1 = 61.205\%$ <br> $C_2 = 76.660\%$ <br> $C_3 = 88.935\%$ | $y_1 = 0.694$ <br> $y_2 = 1.575$ <br> $y_3 = 2.7$ | $y_1 = 0.7$ <br> $y_2 = 1.6$ <br> $y_3 = 2.7$ | $\Delta t_1 = 0.7$ <br> $\Delta t_2 = 0.9$ <br> $\Delta t_3 = 1.1$ |
| **4 FGM layers** | $C_1 = 58.796\ \%$ <br> $C_2 = 71.073\%$ <br> $C_3 = 81.424\%$ <br> $C_4 = 90.140\%$ | $y_1 = 0.505$ <br> $y_2 = 1.107$ <br> $y_3 = 1.830$ <br> $y_4 = 2.7$ | $y_1 = 0.5$ <br> $y_2 = 1.1$ <br> $y_3 = 1.8$ <br> $y_4 = 2.7$ | $\Delta t_1 = 0.5$ <br> $\Delta t_2 = 0.6$ <br> $\Delta t_3 = 0.7$ <br> $\Delta t_4 = 0.9$ |

**Table 6.** Linearized stress in the steel heat sink with a discrteized FGM. Paths are indicated in Figure 5.

| Path | $\overline{P_m + Q_m}$ [MPa] | $\overline{P_m + P_b}$ [MPa] | $\overline{Q}$ [MPa] |
|------|------|------|------|
| **1 FGM layer** | | | |
| $\overline{AB}$ | 15.542 | 10.153 | 85.571 |
| $\overline{CD}$ | 119.71 | 14.533 | 169.06 |
| $\overline{EF}$ | 58.240 | 0.86552 | 137.61 |
| $\overline{GH}$ | 47.237 | 9.6263 | 52.204 |
| **2 FGM layers** | | | |
| $\overline{AB}$ | 15.503 | 10.136 | 85.378 |
| $\overline{CD}$ | 120.11 | 14.515 | 169.07 |
| $\overline{EF}$ | 58.96 | 0.81780 | 138.23 |
| $\overline{GH}$ | 47.204 | 9.6330 | 52.189 |
| **3 FGM layers** | | | |
| $\overline{AB}$ | 15.702 | 10.131 | 85.557 |
| $\overline{CD}$ | 119.95 | 14.516 | 168.78 |
| $\overline{EF}$ | 58.742 | 0.80452 | 138.00 |
| $\overline{GH}$ | 46.984 | 9.6333 | 51.997 |
| **4 FGM layers** | | | |
| $\overline{AB}$ | 15.359 | 10.133 | 85.183 |
| $\overline{CD}$ | 120.43 | 14.516 | 169.28 |
| $\overline{EF}$ | 59.374 | 0.81060 | 138.61 |
| $\overline{GH}$ | 47.345 | 9.6332 | 52.312 |

As a sanity check of the proposed design methodology one carries out a last simulation having the heat sink, which was verified to be able to withstand the loads due to the thermal gradient, directly bonded to the tungsten armor without any interlayer. Therefore the tungsten slab, with thickness $\delta_1 = 2$ mm, is for this case joined directly to the steel heat sink having the same geometry used up to now. Table 7 shows the linearized stresses in the AISI316L in this latter case. By comparing such values with the ones of the four FGM layers in Table 6, one concludes that with the engineered interlayer:

- The linearized secondary stress $\overline{Q}$ is reduced on average by more than 26%. The maximum reduction is of 48%, on the segment $\overline{GH}$, while the minimum reduction is of 0.6%, on the segment $\overline{AB}$. Most importantly, the most loaded section $\overline{CD}$ has a reduction of more than 24% in the secondary stresses.
- The equivalent primary plus secondary membrane stress $\overline{P_m + Q_m}$ is reduced on average by more than 40%. The maximum reduction is of 52 %, on the segment $\overline{EF}$, while the minimum reduction is of 28%, on the segment $\overline{CD}$.

The interlayer designed with the proposed methodology therefore effectively fulfills its purpose of substantially decreasing the thermal stresses inside the heat sink, almost matching the limit case without both the interlayer and the PFM (Table 2). Moreover, if the verification of the design criteria is carried out, shown in Table 8, it is possible to assess that for this component the criteria on the primary plus secondary membrane stresses is not anymore satisfied along the segment $\overline{CD}$, proving the effectiveness of this design study.

**Table 7.** Linearized stress in the steel heat sink having the armor directly joined (no interlayer). Paths are indicated in Figure 5.

| | Armor Directly Joined to the Heat Sink (No Interlayer) | | |
|---|---|---|---|
| Path | $\overline{P_m + Q_m}$ [MPa] | $\overline{P_m + P_b}$ [MPa] | $\overline{Q}$ [MPa] |
| $\overline{AB}$ | 22.875 | 10.030 | 85.752 |
| $\overline{CD}$ | 169.52 | 18.821 | 223.31 |
| $\overline{EF}$ | 125.10 | 2.0706 | 201.74 |
| $\overline{GH}$ | 91.913 | 13.569 | 101.70 |

**Table 8.** Verification of the design criteria on the most critical sections. Case with the armor directly bonded to the heat sink. $T_{avg}$ is the average temperature along the path; $S_m$ is the allowable membrane stress; $S_e$ is the allowable stress intensity. Conservatively $K_{eff}$ is taken as 1. Paths are indicated in Figure 5. On the line segment $\overline{CD}$ a design criterion is not anymore satisfied.

| Path | $T_{avg}$ [°C] | $S_m$ [MPa] | $S_e$ [MPa] | $\dfrac{\overline{P_m + Q_m}}{S_e}$ | $\dfrac{\overline{P_m + P_b}}{K_{eff} S_m}$ | $\dfrac{\overline{P_m + P_b + Q}}{3 S_m}$ |
|---|---|---|---|---|---|---|
| $\overline{AB}$ | 72.034 | 127 | 146.05 | 0.156 | 0.0789 | 0.2513 |
| $\overline{CD}$ | 95.367 | 127 | 138.50 | **1.223** | 0.1481 | 0.6355 |
| $\overline{EF}$ | 88.458 | 127 | 140.75 | 0.8888 | 0.0163 | 0.5349 |
| $\overline{GH}$ | 90.707 | 127 | 140.02 | 0.6564 | 0.1068 | 0.3025 |

## 4. Discussion and Conclusions

In this paper we presented a comprehensive study on how to reduce the thermal stresses inside the PFCs, using a FGM interlayer to be interposed between the armor and the heat sink. The main design rationale is to develop the interlayer so that it ensures the kinematic continuity among all the sub-components, in a configuration where they deform into exactly the shape they would assume if they were detached from each other. Therefore, we first computed the shape that isostatically constrained bodies with geometries relevant for PFCs would assume only due the effect of a temperature field arising along their thickness. Such expansion is, under the assumption of thin solids, tightly linked to the mean value of the thermal strain field.

When no bending occurs, as in the case of a thin hollow cylinder, we show that the thermal stresses due to the CTE mismatch are zero when the mean thermal strain of the armor and the heat sink is the same. Therefore, the first goal of the interlayer is in this case to bring the PFM up to a suitable temperature field such that it will expand as much as the heat sink. This does not necessarily imply the use of a FGM, also a poorly conductive interlayer (e.g., thermal break [25]) can be design following our design rationale. Moreover, a poorly conductive armor material could be beneficial to the thermal stresses in the PFC, as long as the loads due to the thermal gradient inside the PFM, which are worsened by a low thermal conductivity, are below the allowable stress. In any case, the thermal stresses in the heat sink would always benefit from a hotter W armor, since the matching of the thermal expansion would be improved. The major benefit of a FGM interlayer, compared to other solutions, is that its W concentration can be tailored to ideally eliminate the stresses inside itself.

When bending occurs, as in the case of the plate, the proper matching of the shape of both the armor and the heat sink is achieved when the curvature that they would have if detached from each other is the same. We showed that such curvature depends only on the heat flux, which is imposed by the plasma, and on the material properties, specifically the ratio $\frac{\alpha}{\lambda}$. Therefore, a solution exists only if the PFM and the heat sink material have the same $\frac{\alpha}{\lambda}$. Unfortunately, this is not verified for the current candidates of PFC materials. Therefore,

the total strain field of the armor and heat sink cannot be paired, at least generally, as was performed in the previous case.

In the design study presented in this paper, we hindered the bending of the component by imposing an external kinematic condition. It will then be necessary that a proper fixation system is designed to best achieve this condition in a real component. The design methodology we propose is a direct application of the procedure developed in the analytical cases. However, a number of approximations has been performed to take into account the deviations that the realistic study have from the ideal cases. The interlayer thickness is determined in a way such that the PFM is hot enough that its mean thermal strain is equal to the target value of the total strain (that we call $\varepsilon_0$), while its ideal concentration is computed such that the thermal strain of this sub-component is constant and equal to $\varepsilon_0$. In this study, the target value of the total strain has been obtained as the mean value of the heat sink total strain, computed with a FEM simulation. The ideal concentration function $C$ of the FGM was then discretized in layers by imposing that the stresses are distributed evenly among the layers. To the best knowledge of the authors, up to now, different trends of $C$ were either tested in a "trial-and-error" fashion, in order to find the one corresponding to the best configuration [10], or found by solving numerically an optimization problem [11–13]. In this paper we attempted to determine the concentration function of a FGM interlayer analytically, starting from some thermomechanical criteria.

From the results of the design study carried out applying the proposed methodology, we derive the following conclusions:

- The proposed methodology allows for an effective design of a PFC having a rather insulating heat sink material, the AISI 316L ($\lambda_{\text{steel}} \sim 15 \, \text{Wm}^{-1}\text{K}^{-1}$).
- For the FGM concentration function, the traditionally fixed constraints [10,20] $C = 0$ at the interface with steel and $C = 1$ at the boundary with the PFM are proved to have a negative impact on the thermal stresses, since they lead to a mismatch of "free-to-deform" shape of the sub-components.
- The number of FGM layers do not impact the stresses in both the armor and the heat sink, due to the design methodology. The discretization depends only on the desired maximum stress in the FGM due to the thermal gradient, which decreases with a higher amount of layers. The case with an ideal FGM represents its minimum limit.
- The engineered interlayer proved to reduce significantly the secondary stress in the PFC, compared to a case with the armor directly bonded to the heat sink. The linearized secondary stress $\overline{Q}$ is reduced on average by more than 26%. Most importantly, the most loaded section has a reduction of more than 24% in the secondary stresses. The equivalent primary plus secondary membrane stress $\overline{P_m + Q_m}$ is reduced on average by more than 40%, with a reduction of 28% on the most critical segment of the heat sink.

**Author Contributions:** Conceptualization, S.R.; Methodology, G.D.; Supervision, F.R.; Writing—original draft, G.D. and S.R.; Writing—review & editing, F.R. All authors have read and agreed to the published version of the manuscript.

**Funding:** This research received no external funding.

**Informed Consent Statement:** Not applicable.

**Data Availability Statement:** Not applicable.

**Conflicts of Interest:** The authors declare no conflict of interest.

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
