# Peer review of "Engineering of a FGM Interlayer to Reduce the Thermal Stresses Inside the PFCs"

_applsci, doi:10.3390/app122010215_

Round 1

Reviewer 1 Report

In your manuscript you have provided a detailed model for a construction of FGM for fusion application.

I would recommend minor changes, namely I am missing a bit more of a connection to the existing FGMs. So, I would recommend to add/to discuss how your model connects to the already produced materials.

Author Response

Dear Reviewer,

Thank you for your comments.

My co-authors and I have added a small paragraph expanding on the relation between the modelling of the FGM and the FGM already produced, with the corresponding references.

You will find such a part in the lines 143-149 of the new manuscript, whose pdf I attach to this response.

Thank you again,

Best regards,

Giacomo Dose

Reviewer 2 Report

In my opinion, the manuscript is appropriate for publication.

Author Response

Dear Reviewer,

Thank you for the revison.

Best Regards

Reviewer 3 Report

This paper is suitable to publish in this journal. It is well written and has novel results.

Author Response

(The authors gave the same response as above.)

Reviewer 4 Report

The authors present an analytical work about the functionally graded material's design to reduce thermal stress in the plasma-facing components. This study provides several practical methodologies to achieve the analysis by simulation and conclude several applicable criteria for minimizing thermal stress. I believe this manuscript is of interest to those working in the field of plasma-facing components in a fusion reactor. Furthermore, the structure of this manuscript is well organized, and the results are clearly presented. Therefore, I recommend this manuscript be accepted in its present form. 

Author Response

(The authors gave the same response as above.)
